# An Analytical Solution for Longitudinal Impedance of a Large-Diameter Floating Pile in Soil with Radial Heterogeneity and Viscous-Type Damping

**Kun Meng [1,2], Chunyi Cui [1,*], Zhimeng Liang [1], Haijiang Li [2] and Huafu Pei [3]**

[1]  Department of Civil Engineering, Dalian Maritime University, Dalian 116026, China;
    mengkuntumu@dlmu.edu.cn (K.M.); liangzhimeng@dlmu.edu.cn (Z.L.)
[2]  Cardiff School of Engineering, Cardiff University, Queen's Buildings, The Parade, Cardiff CF24 3AA, UK;
    LiH@cardiff.ac.uk
[3]  Faculty of Infrastructure Engineering, Dalian University of Technology, Dalian 116024, China;
    huafupei@dlut.edu.cn
*   Correspondence: cuichunyi@dlmu.edu.cn; Tel.: +86-411-84723186

**Abstract:** An analytical model is presented for solving the longitudinal complex impedance of a large-diameter floating pile in viscoelastic surrounding soil with radial heterogeneity and viscous-type damping, taking the effect of three-dimensional wave propagation of soil and lateral inertia of the pile shaft into account. The corresponding analytical solution for longitudinal impedance is also derived and validated via comparisons with existing solutions. The influences of the pile length, Poisson's ratio of the pile shaft and the viscous damping coefficient, as well as the degree and radius of disturbed surrounding soil, on the longitudinal impedance of the pile shaft are examined by performing parametric analyses. It is demonstrated that the proposed analytical model and solution are suitable for the longitudinal vibration problem of a large-diameter pile and radially inhomogeneous surrounding soil, especially when the pile slenderness is low. In addition, the present solution can be easily degenerated to describe the longitudinal vibration problem relating to a large-diameter floating pile in radially homogenous soil or a pile with fixed-end supports.

**Keywords:** pile vibration; longitudinal impedance; analytical solution; radial heterogeneity; viscous-type damping

## 1. Introduction

In most analytical models for pile vibration, the soil around the pile shaft is generally simplified to be radially homogeneous [1–6]. However, when construction operations cause soil to be disturbed within the vicinity of the pile, the effect of the soil's radial heterogeneity on the vibration performance of the pile–soil system cannot be roughly ignored. In recent decades, the vibration problems of piles in radial inhomogeneous soil have been focused on by researchers [7,8]. As pioneering work, Novak et al. [9,10] and Veletsos et al. [11,12] investigated the longitudinal and torsional vibration of piles in radially heterogeneous soil by dividing the surrounding soil into a semi-infinite outer undisturbed zone and an inner disturbed zone with a single layer. Subsequently, Nogami et al. [13,14] derived an analytical solution for the longitudinal impedance of piles in radially inhomogeneous soil by combining Novak's planestrain and Winkler's models. Doston et al. [15] deduced the analytical expressions for both the longitudinal and torsional impedance of a single pile in soil with radial heterogeneity, assuming an exponential function for the variance of soil shear moduli within the inner zone. Similarly, Vaziri et al. [16] and Han et al. [17,18] considered the variance of shear moduli as a parabolic function for the radial heterogeneity of surrounding soil. Furthermore, EI Naggar [19,20] regarded the

inner disturbed zone of surrounding soil as a series of annular sub-layers and combined them with Novak's plane-strain model for pile–soil vibration. On this basis, Wang et al. [21] and Yang et al. [22] identified the limitations of EI Naggar's model and proposed a new approach for the longitudinal vibration of piles in hysteretic-damping soil, which considers a completely coupled condition at the sub-layer interface by adopting the complex stiffness method. In addition, Dai et al. [23] further examined the longitudinal impedance of piles in radially inhomogeneous soil with a hysteretic-type damping model, considering the three-dimensional (3D) wave propagation in soil.

The aforementioned studies mainly employed the Euler–Bernoulli rod model to describe the dynamic behavior of pile shafts, in which the wave propagation effect of pile shafts in a radial direction was roughly ignored [24,25]. Instead, the Rayleigh–Love rod model can take account of this radial wave effect, i.e., the lateral inertia effect, by introducing Poisson's ratio into the governing equation [26–28]. With the combination of the Rayleigh–Love model and the 3D wave propagation theory, Lü et al. [29,30] proposed a simplified model for the longitudinal dynamic behavior of a large-diameter pile in radially homogenous viscoelastic media by adopting the hysteretic-type damping model. Afterwards, Zheng et al. [31] further extended this model to examine the longitudinal vibration of a large-diameter pipe pile in radially homogenous media. Moreover, for the longitudinal vibration problem of piles with radial heterogeneity, Li et al. [32,33] investigated the longitudinal vibration characteristics of a large-diameter pile in viscoelastic media with radial heterogeneity and hysteretic-type damping, using Novak's thin layers model and the wave propagation theory of a 3D continuum, respectively. It has been demonstrated that the hysteretic-type damping model, which is independent of frequency, could be unsatisfactory when the excitation is non-harmonic [34]. In contrast, the viscous-type damping model is suitable for non-harmonic excitation [35,36]. Hence, Cui et al. [37,38] presented a new mechanical model and examined the longitudinal impedance of a pipe pile in layered viscoelastic media with radial heterogeneity and viscous-type damping, based on Novak's plane-strain model.

To date, however, little work has been carried out on the longitudinal impedance of a large-diameter floating pile in viscoelastic soil with radial heterogeneity, combining both the viscous-type damping and the 3D wave propagation effect of surrounding soil. The primary aim of this paper is to develop a new analytical model to describe the longitudinal vibration of a large-diameter floating pile in viscoelastic surrounding soil with radial inhomogeneity, taking the effect of 3D wave propagation and lateral inertia of the pile shaft into account. In addition, extensive parametric analyses are also conducted to examine the longitudinal vibration characteristics of floating piles in surrounding soil with radial heterogeneity.

## 2. Computational Model and Basic Assumptions

Figure 1 shows a new mechanical model for the longitudinal vibration of an interaction system including soil and a solid pile. $H$, $r_1$, $\rho^p$, $E^p$ and $\nu^p$ represent the length, diameter, density, elastic modulus and Poisson's ratio, respectively, of a floating solid pile. The surrounding soil is composed of two parts, i.e., two zones: an inner disturbed annular zone and a semi-infinite outer undisturbed zone of surrounding soil. The inner zone of surrounding soil can be further divided into $m$ annular sub-layers. The radial thickness and radius of the inner disturbed zone are $b$ and $r_{m+1}$, respectively. $r_{j+1}$ is the outer radius of the $j$th annular sub-layer. $\rho_j^s$, $\lambda_j^s$, $G_j^s$, $E_j^s$ and $c_j^s$ denote the density, Lame constant, shear modulus, elastic modulus and viscous coefficient, respectively, of the $j$th disturbed sub-layer. The mechanical constants of the viscoelastic supports beneath the pile toe and surrounding soil are $k^p$, $\delta^p$ and $k^s$, $\delta^s$, respectively. The uniformly distributed excitation pressure is $p(t)$. Besides this, the following assumptions are specified in the proposed mechanical model:

(1)　The large-diameter pile is considered to be a Rayleigh–Love rod with linear elasticity and a uniform cross section, while the soil is an isotropic viscoelastic continuum with frequency-dependent viscous-type damping [38].

(2)　Within the inner zone of surrounding soil, two neighboring annular sub-layers are completely coupled at the interface.

(3)　The deformation of the simplified mechanical system is small. There is no interface sliding between the pile and soil.

(4)　Within the inner zone of surrounding soil, $G_j^s$ and $c_j^s$ are determined in terms of the following expressions:

$$G_j^s(r) = \begin{cases} G_1^s & r = r_1 \\ G_{m+1}^s \times f(r) & r_1 < r < r_{m+1} \\ G_{m+1}^s & r \geq r_{m+1} \end{cases} \tag{1a}$$

$$c_j^s(r) = \begin{cases} c_1^s & r = r_1 \\ c_{m+1}^s \times f(r) & r_1 < r < r_{m+1} \\ c_{m+1}^s & r \geq r_{m+1} \end{cases} \tag{1b}$$

where $f(r)$ is a parabolic function that describes the variance of construction disturbance for the *j*th disturbed sub-layer [22].

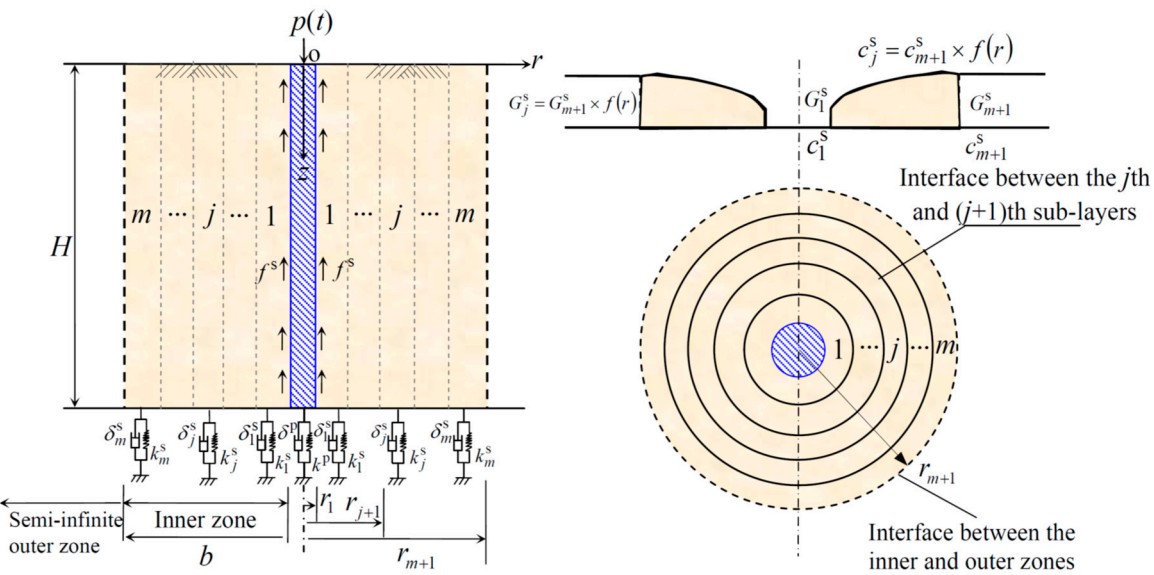

**Figure 1.** The simplified mechanical model.

The mathematical implementation for the derivation procedure of pile–soil dynamic vibration is shown in Figure 2.

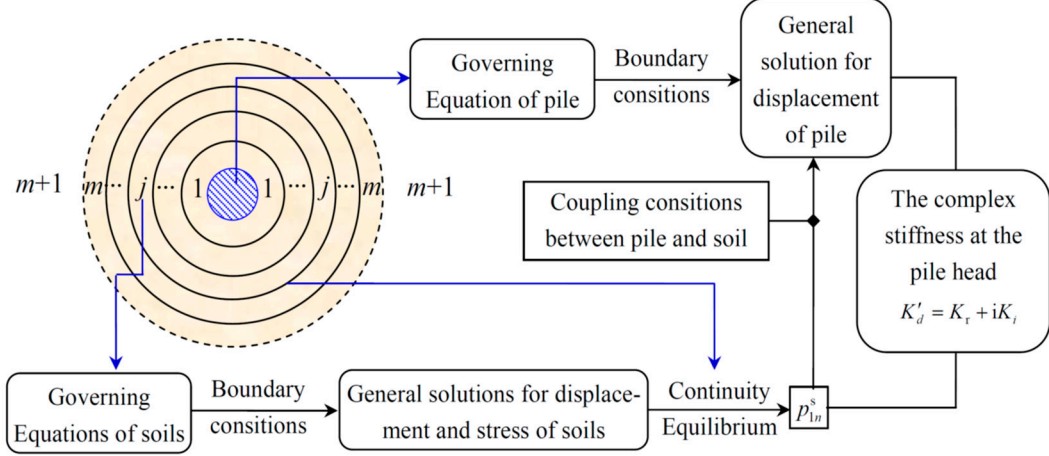

**Figure 2.** The derivation procedure of complex stiffness.

## 3. Governing Equations

Based on the wave propagation theory of a continuum in the axisymmetric condition, the governing equation for the *j*th disturbed sub-layer proposed by Nogami and Novak [34] is adopted:

$$(\lambda_j^s + 2G_j^s)\frac{\partial^2}{\partial z^2}u_j^s(r,z,t) + G_j^s(\frac{1}{r}\frac{\partial}{\partial r} + \frac{\partial^2}{\partial r^2})u_j^s(r,z,t) + c_j^s\frac{\partial}{\partial t}\left[\left(\frac{\partial^2}{\partial z^2} + \frac{1}{r}\frac{\partial}{\partial r} + \frac{\partial^2}{\partial r^2}\right)u_j^s(r,z,t)\right] = \rho_j^s\frac{\partial^2}{\partial t^2}u_j^s(r,z,t) \tag{2}$$

where $u_j^s(r,z,t)$ is the longitudinal displacement of the *j*th disturbed sub-layer.

The longitudinal shear stress at the interface (i.e., $r = r_1$) between the first annular sub-layer and the pile shaft is expressed by

$$\tau_1^s(r_1,z,t) = G_1^s\frac{\partial u_1^s(r_1,z,t)}{\partial r} + c_1^s\frac{\partial^2 u_1^s(r_1,z,t)}{\partial t \partial r}. \tag{3}$$

According to the Rayleigh–Love rod model theory [31], the governing equation for the longitudinal vibration of a large-diameter pile shaft can be written as

$$E^P A^P\frac{\partial^2 u^P(z,t)}{\partial z^2} - m^P\left(\frac{\partial^2 u^P(z,t)}{\partial t^2} + (v^P r_1)^2\frac{\partial^4 u^P(z,t)}{\partial z^2 \partial t^2}\right) - 2\pi r_1 f^s(z,t) = 0 \tag{4}$$

where $f^s(z,t) = \tau_1^s(r,z,t)\big|_{r=r_1}$, $m^P = \rho^P A^P$, $A^P = \pi r_1^2$.

## 4. Boundary and Initial Conditions

The boundary conditions at the free surface and bottom of the *j*th annular sub-layer are given by the following expressions, respectively.

$$\frac{\partial u_j^s(r,z,t)}{\partial z}\big|_{z=0} = 0 \tag{5a}$$

$$\frac{\partial u_j^s(r,z,t)}{\partial z}\big|_{z=H} = -\left(\frac{k_j^s u_j^s(r,z,t)}{E_j^s} + \frac{\delta_j^s}{E_j^s}\frac{\partial u_j^s(r,z,t)}{\partial t}\right) \tag{5b}$$

where $E_j^s\frac{\partial u_j^s(r,z,t)}{\partial z}$ refers to the internal stress of the soil. $k_j^s u_j^s(r,z,t)$ and $\delta_j^s\frac{\partial u_j^s(r,z,t)}{\partial t}$ are the external forces related to stiffness and damping, respectively. According to the force equilibrium conditions, Equation (5b) can be established.

When $r \to \infty$, the longitudinal displacement of surrounding soil tends toward zero. Namely,

$$\lim_{r \to \infty} u(r, z, t) = 0 \tag{6}$$

The continuity and equilibrium conditions at the outer interface of the *j*th sub-layer are

$$u_j^s(r, z, t)\big|_{r=r_{j+1}} = u_{j+1}^s(r, z, t)\big|_{r=r_{j+1}} \tag{7a}$$

$$G_j^s \frac{\partial u_j^s(r, z, t)}{\partial r} + c_j^s \frac{\partial^2 u_j^s(r, z, t)}{\partial t \partial r}\bigg|_{r=r_{j+1}} = G_{j+1}^s \frac{\partial u_{j+1}^s(r, z, t)}{\partial r} + c_{j+1}^s \frac{\partial^2 u_{j+1}^s(r, z, t)}{\partial t \partial r}\bigg|_{r=r_{j+1}} \tag{7b}$$

where $G_j^s \frac{\partial u_j^s(r,z,t)}{\partial r} + c_j^s \frac{\partial^2 u_j^s(r,z,t)}{\partial t \partial r}\big|_{r=r_{j+1}}$ and $G_{j+1}^s \frac{\partial u_{j+1}^s(r,z,t)}{\partial r} + c_{j+1}^s \frac{\partial^2 u_{j+1}^s(r,z,t)}{\partial t \partial r}\big|_{r=r_{j+1}}$ are the shear stress of the *j*th and (*j*+1)th, respectively, sub-layers. Based on the stress equilibrium condition at the interface between the *j*th and (*j*+1) th sub-layers, Equation (7b) is built.

The equilibrium conditions of the pile are expressed as the following forms:

$$E^P A^P \frac{du^P}{dz}\bigg|_{Z=0} + \rho^P A^P (v^P r_1 s)^2 \frac{du^P}{dz}\big|_{z=0} = -p(t) \tag{8a}$$

$$\frac{k^P + s\delta^P}{E^P A^P} u^P + \frac{du^P}{dz} + \frac{\rho^P A^P (v^P r_1 s)^2}{E^P A^P} \frac{du^P}{dz}\big|_{z=H} = 0 \tag{8b}$$

where $\frac{k^P + s\delta^P}{E^P A^P} u^P$ denotes the external supporting force beneath the pile toe. $\frac{du^P}{dz} + \frac{\rho^P A^P (v^P r_1 s)^2}{E^P A^P} \frac{du^P}{dz}$ is the internal stress of the pile. According to the force equilibrium condition beneath the pile toe, Equation (8b) can be established.

The coupled condition of the first disturbed sub-layer and pile is given by

$$u_1^s(r, z, t)\big|_{r=r_1} = u^P(z, t). \tag{9}$$

## 5. Solution of the Surrounding Soil

Performing a Laplace transform to Equation (2), it gives

$$(\lambda_j^s + 2G_j^s)\frac{\partial^2}{\partial z^2} U_j^s(r,z,s) + G_j^s(\frac{1}{r}\frac{\partial}{\partial r} + \frac{\partial^2}{\partial r^2})U_j^s(r,z,s) + c_j^s s\left(\frac{\partial^2}{\partial z^2} + \frac{1}{r}\frac{\partial}{\partial r} + \frac{\partial^2}{\partial r^2}\right)U_j^s(r,z,s) = \rho_j^s s^2 U_j^s(r,z,s) \tag{10}$$

where $U_j^s(r, z, s)$ is the Laplace transform of $u_j^s(r, z, t)$.

Setting $U_j^s(r, z, s) = R_j^s(r)Z_j^s(z)$ and substituting it into Equation (10) with rearrangement yields

$$(\lambda_j^s + 2G_j^s + c_j^s s)\frac{1}{Z_j^s}\frac{\partial^2 Z_j^s}{\partial z^2} - \rho_j^s s^2 + (G_j^s + c_j^s s)\frac{1}{R_j^s}\left(\frac{1}{r}\frac{\partial R_j^s}{\partial r} + \frac{\partial^2 R_j^s}{\partial r^2}\right) = 0. \tag{11}$$

Further splitting Equation (11), the following ordinary differential equations can be given:

$$\frac{d^2 Z_j^s}{dz^2} + (h_j^s)^2 Z_j^s = 0 \tag{12a}$$

$$\frac{d^2 R_j^s}{dr^2} + \frac{1}{r}\frac{dR_j^s}{dr} - (q_j^s)^2 R_j^s = 0 \tag{12b}$$

where $h_j^s$ and $q_j^s$ are undetermined coefficients that satisfy the following expression:

$$-(\lambda_j^s + 2G_j^s + c_j^s s)(h_j^s)^2 + (G_j^s + c_j^s s)(q_j^s)^2 = \rho_j^s s^2. \tag{13}$$

Rearranging the terms of Equation (13) gives

$$(q_j^s)^2 = \frac{(\lambda_j^s + 2G_j^s + c_j^s s)(h_j^s)^2 + \rho_j^s s^2}{(G_j^s + c_j^s s)}. \tag{14}$$

Thus, the general solutions of Equations (12a) and (12b) are

$$Z_j^s(z) = C_j^s \cos(h_j^s z) + D_j^s \sin(h_j^s z) \tag{15a}$$

$$R_j^s(r) = A_j^s I_0(q_j^s r) + B_j^s K_0(q_j^s r) \tag{15b}$$

where $I_0(q_j^s r)$ and $K_0(q_j^s r)$ are the first and second kind, respectively, modified Bessel functions of order zero; $A_j^s$, $B_j^s$, $C_j^s$ and $D_j^s$ are undetermined coefficients.

Substituting $U_j^s(r, z, s) = R_j^s(r) Z_j^s(z)$ into Equations (5a) and (5b), respectively, gives

$$D_j^s = 0 \tag{16a}$$

$$\tan(h_j^s H) = \frac{\overline{K}_j^s}{h_j^s H} \tag{16b}$$

where $\overline{K}_j^s = K_j^s H / E_j^s$; $K_j^s = k_j^s + s\delta_j^s$ denotes the complex stiffness of viscoelastic supports beneath the pile toe.

Solving the transcendental Equation (16b) yields the eigenvalues $h_{jn}^s$ ($n=1, 2, \dots$ ). Then, $q_{jn}^s$ can be further obtained with the substitution of $h_{jn}^s$ into Equation (14).

Combining Equations (5a), (5b) and (6), the general solution $U_j^s$ can be obtained as

$$U_j^s = \begin{cases} \sum\limits_{n=1}^{\infty} A_{jn}^s K_0(q_{jn}^s r) \cos(h_{jn}^s z) & (j = m+1) \\ \sum\limits_{n=1}^{\infty} \left[ B_{jn}^s I_0(q_{jn}^s r) + C_{jn}^s K_0(q_{jn}^s r) \right] \cos(h_{jn}^s z) & (j = m, \dots, 2, 1) \end{cases} \tag{17}$$

where $A_{jn}^s$, $B_{jn}^s$ and $C_{jn}^s$ are undetermined coefficients.

Hence, the shear stress at the inner interface of the $j$th sub-layer can be further expressed as

$$\tau_j^s = \begin{cases} (G_j^s + c_j^s s) \sum\limits_{n=1}^{\infty} A_{jn}^s q_{jn}^s K_1(q_{jn}^s r) \cos(h_{1jn}^s z), & (j = m+1) \\ (G_j^s + c_j^s s) \sum\limits_{n=1}^{\infty} q_{jn}^s \left[ -B_{jn}^s I_1(q_{jn}^s r) + C_{jn}^s K_1(q_{jn}^s r) \right] \cos(h_{jn}^s z), & (j = m, \dots, 2, 1) \end{cases}. \tag{18}$$

Considering the boundary conditions listed in Equations (7a) and (7b), $p_{jn}^s = B_{jn}^s / C_{jn}^s$ is obtained as

$$p_{mn}^s = \frac{(G_m^s + c_m^s s) q_{mn}^s K_1(q_{mn}^s r_m) K_0(q_{(m+1)n}^s r_m) - (G_{m+1}^s + c_{m+1}^s s) K_0(q_{mn}^s r_m) K_1(q_{(m+1)n}^s r_m)}{(G_m^s + c_m^s s) q_{mn}^s I_1(q_{mn}^s r_m) K_0(q_{(m+1)n}^s r_m) + (G_{m+1}^s + c_{m+1}^s s) q_{(m+1)n}^s I_0(q_{mn}^s r_m) K_1(q_{(m+1)n}^s r_m)} \quad (j = m) \tag{19a}$$

$$p_{jn}^s = \frac{\begin{aligned} &(G_j^s + c_j^s s) q_{jn}^s K_1(q_{jn}^s r_j) [q_{(j+1)n}^s I_0(q_{(j+1)n}^s r_j) + K_0(q_{(j+1)n}^s r_j)] \\ &- (G_{j+1}^s + c_{j+1}^s s) q_{(j+1)n}^s K_0(q_{jn}^s r j) [q_{(j+1)n}^s I_1(q_{(j+1)n}^s r_j) - K_1(q_{(j+1)n}^s r_j)] \end{aligned}}{\begin{aligned} &(G_j^s + c_j^s s) q_{jn}^s I_1(q_{jn}^s r_j) [q_{(j+1)n}^s I_0(q_{(j+1)n}^s r_j) + K_0(q_{(j+1)n}^s r_j)] \\ &- (G_{j+1}^s + c_{j+1}^s s) q_{(j+1)n}^s I_0(q_{jn}^s r_j) [q_{(j+1)n}^s I_1(q_{(j+1)n}^s r_j) - K_1(q_{(j+1)n}^s r_j)] \end{aligned}} \quad (j = m-1, \dots, 2, 1). \tag{19b}$$

## 6. Solution of the Large-Diameter Pile

Substituting Equation (3) into Equation (4) and applying a Laplace transform produces

$$[(V^P)^2 + (v^P r s)^2] \frac{\partial^2 U_1^P(z,s)}{\partial z^2} - s^2 U^P(z,s) - \frac{2\pi r_1}{\rho^P A^P}(G_1^s + c_1^s s) \sum_{n=1}^{\infty} q_{1n}^s \left\{[-B_{1n}^s I_1(q_{1n}^s r_1) + C_{1n}^s K_1(q_{1n}^s r)]\cos(h_{1n}^s z)\right\} = 0 \qquad (20)$$

where $V^P = \sqrt{E^P/\rho^P}$, and $U^P(z,s)$ is the Laplace transform of $u^P(z,t)$.

Setting $s = i\omega(i = \sqrt{-1})$, the general solution for Equation (20) is achieved as

$$U^{P\prime} = D_1^P \cos(\frac{\omega}{\eta}z) + D_2^P \sin(\frac{\omega}{\eta}z). \qquad (21)$$

Furthermore, the particular solution for Equation (20) is given by

$$U^{P*} = \sum_{n=1}^{\infty} M_n^s \cos(h_{1n}^s z) \qquad (22)$$

where $D_1^P$, $D_2^P$ and $M_n^s$ are undetermined coefficients; $\eta = \sqrt{(V^P)^2 + (v^P r s)^2}$.

Substituting Equation (22) into Equation (20) with rearrangement yields

$$M_n^s = \frac{2\pi r_1 q_{1n}^s}{\rho^P A^P} \frac{(G_1^s + c_1^s s)[B_{1n}^s I_1(q_{1n}^s r_1) - C_{1n}^s K_1(q_{1n}^s r_1)]}{[(V^P)^2 + (v^P r_1 s)^2](h_{1n}^s)^2 - \omega^2}. \qquad (23)$$

Therefore, the solution for Equation (21) is obtained as

$$U^P = D_1^P \cos(\frac{\omega}{\eta}z) + D_2^P \sin(\frac{\omega}{\eta}z) + \sum_{n=1}^{\infty} M_n^s \cos(h_{1n}^s z). \qquad (24)$$

Combining Equations (9), (19a), (19b) adn (24) gives

$$U^P = D_1^P[\cos(\frac{\omega}{\eta}z) + \sum_{n=1}^{\infty} \gamma_n' \cos(h_{1n}^s z)] + D_2^P[\sin(\frac{\omega}{\eta}z) - \sum_{n=1}^{\infty} \gamma_n'' \cos(h_{1n}^s z)] \qquad (25)$$

where the coefficients of $\gamma_n'$ and $\gamma_n''$ are provided in Appendix A.

In terms of Equations (8a) and (8b), the longitudinal impedance at the head of the pile shaft can be expressed by

$$Z(\theta) = \frac{P(s)}{U^P(z,s)} = \frac{E^P A^P}{H} \frac{[1 - (v^P \overline{r_1} \theta \eta)^2/(V^P)^2]\theta}{\frac{D_1^P}{D_2^P}(1 + \sum_{n=1}^{\infty} \gamma_n') - \sum_{n=1}^{\infty} \gamma_n''} = \frac{E^P A^P}{H} K_d' \qquad (26)$$

where $K_d' = \theta[1 - (v^P \overline{r_1}\theta \eta)^2/(V^P)^2]/[\frac{D_1^P}{D_2^P}(1 + \sum_{n=1}^{\infty} \gamma_n') - \sum_{n=1}^{\infty} \gamma_n'']$ is the dimensionless complex stiffness and $P(s)$ is the Laplace transform of $p(t)$.

$K_d'$ can be rewritten as

$$K_d' = K_r + iK_i \qquad (27)$$

where $K_r$ and $K_i$ are the true stiffness and equivalent damping; $R = \frac{k^P}{E^P A^P}H$, $A_b = \frac{\delta^P}{E^P A^P}H$,

$$\frac{D_1^P}{D_2^P} = \frac{\sum_{n=1}^{\infty} \gamma_n'' \sin(\overline{h}_{1n}^s) + (R + sA_b)[\sin(\theta) - \sum_{n=1}^{\infty} \gamma_n'' \cos(\overline{h}_{1n}^s)] + \left(\frac{v^P \overline{r_1}\theta \eta}{V^P}\right)^2 \left[\sum_{n=1}^{\infty} \gamma_n'' \overline{h}_{1n}^s \sin(\overline{h}_{1n}^s) + \theta \cos(\theta)\right] + \theta \cos(\theta)}{\sum_{n=1}^{\infty} \gamma_n' \sin(\overline{h}_{1n}^s) - (R + sA_b)[\cos(\theta) - \sum_{n=1}^{\infty} \gamma_n' \cos(\overline{h}_{1n}^s)] + \left(\frac{v^P \overline{r_1}\theta \eta}{V^P}\right)^2 \left[\sum_{n=1}^{\infty} \gamma_n' \overline{h}_{1n}^s \sin(\overline{h}_{1n}^s) + \theta \sin(\theta)\right] + \theta \sin(\theta)}.$$

## 7. Results and Discussions

Numerical examples are provided to validate the obtained analytical solutions via comparisons with previous solutions. Parametric analyses are also performed to discuss the longitudinal vibration of a large-diameter floating pile embedded in surrounding soil with radial heterogeneity, considering the 3D wave propagation effect. The number of the annular sub-layers $n$ is taken as 20 to satisfy the accuracy requirement in the following analyses, which is suggested in EI Naggar [20] and Cui et al. [38]. Furthermore, a quadratic variation of the shear modulus and viscous damping coefficient, i.e., a linear variation of shear velocity, is assumed as follows:

$$\zeta^s = \sqrt{G_1^s/G_{m+1}^s} = \sqrt{c_1^s/c_{m+1}^s} = V_1^s/V_{m+1}^s \tag{28}$$

where $\zeta^s$ denotes the coefficient of disturbance degree [21]. When $\zeta^s < 1$, the soil is weakened due to construction disturbance; when $\zeta^s > 1$, the soil is strengthened; and when $\zeta^s = 1$, the surrounding soil is radially homogeneous without construction disturbance.

Unless otherwise specified, the following mechanical parameters are used: $r_1 = b = 0.5\text{m}, \rho^p = 2500\text{kg/m}^3, V^p = 4000\text{m/s}, H = 10\text{m}, k^p = 1 \times 10^5 \text{kN/m}^3, \delta^p = 1 \times 10^5 \text{kN} \cdot \text{s/m}^2, v^p = 0.3, v_j^s = 0.25, \rho_j^s = 2000\text{kg/m}^3, V_{m+1}^s = 100\text{m/s}, c_{m+1}^s = 1\text{kN} \cdot \text{s/m}^2, \zeta^s = 1.4$.

### 7.1. Verification of the Solution

With respect to the same parameters, the present solution for a pile head's complex stiffness is reduced to compare with the existing solution of Lü et al. [29] by setting $\zeta^s \to 1$. Figure 3 shows that the present solution for longitudinal impedance with different values of pile length $H$ is in very good agreement with that derived by Lü et al. [29]. Moreover, the material damping of the present solution is viscous-type, which differs from the hysteretic-type damping used for the solution of Yang et al. [22]. For convenience, the effect of material damping is not considered in the following comparison. The present solution is reduced to compare with the existing solution of Yang et al. [22] by setting $c_j^s \to 0$ ($j$=1, 2,..., $m$) and $v^p \to 0$. It is illustrated in Figure 4 that the obtained solution with a different pile length agrees well with the existing solution achieved by Yang et al. [22]. Hence, the accuracy of the present solution can be validated with these independent comparisons.

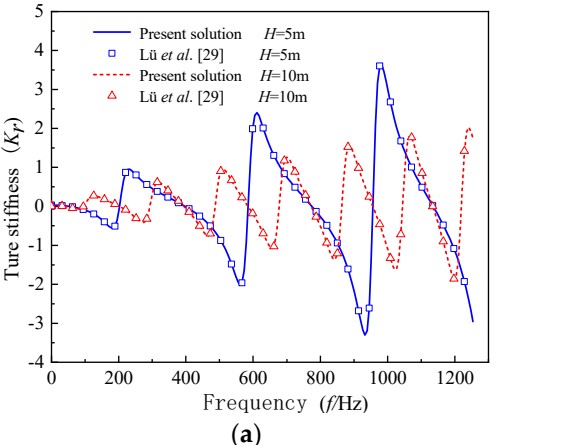
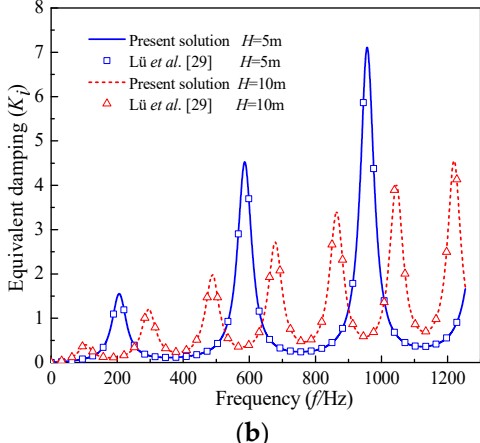

**Figure 3.** Comparison between the present solution ($\zeta^s \to 1$) and the solution of Lü et al. [29]: (**a**) true stiffness; (**b**) equivalent damping.

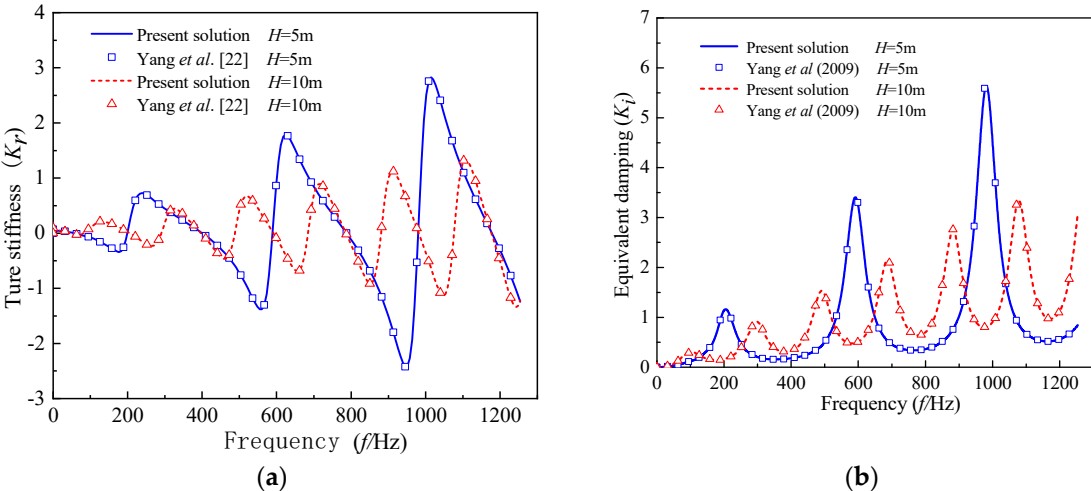

**Figure 4.** Comparison between the present solution ($c_j^s \to 0$ and $v^p \to 0$) and the results of Yang et al. [22]: (**a**) true stiffness; (**b**) equivalent damping.

### 7.2. Parametric Analyses

Due to the consideration of the lateral inertia effect of the pile shaft, the present solution from the Rayleigh–Love rod model can easily be reduced to the one from the Euler–Bernoulli rod model by setting $v^p \to 0$. The effect of Poisson's ratio on the longitudinal impedance at the pile head is shown in Figure 5. It is clear that both the resonance frequency and amplitude of the pile head's longitudinal impedance decline with the increasing Poisson's ratio in the high-frequency range, while the effect of Poisson's ratio on the longitudinal impedance can be neglected in the low-frequency range. Furthermore, it illustrates the limitation of the Euler–Bernoulli rod model ($v^p \to 0$) to describe the longitudinal vibration of a large-diameter pile, compared with the Rayleigh–Love model.

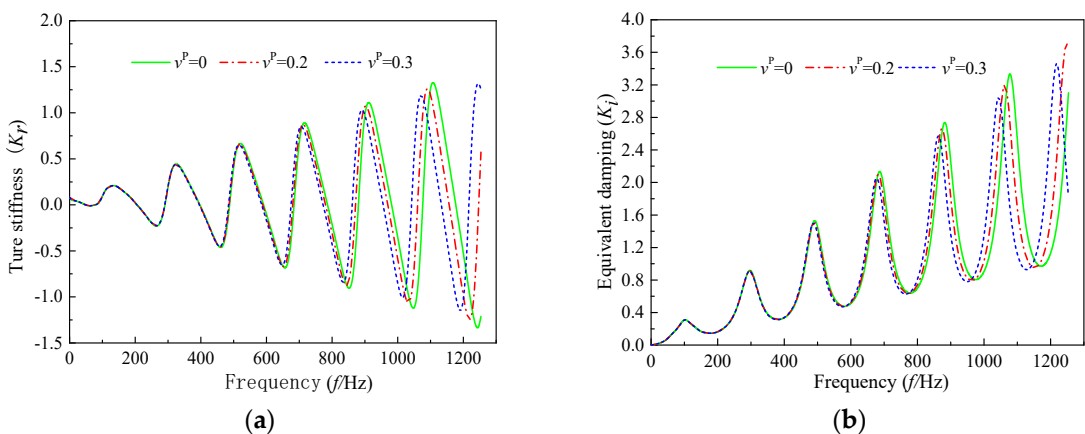

**Figure 5.** The effect of Poisson's ratio of pile shafts on the longitudinal impedance of pile shafts: (**a**) true stiffness; (**b**) equivalent damping.

With the aim to further illustrate the difference between the Rayleigh–Love rod and Euler–Bernoulli models, two cases for Poisson's ratio with $v^p = 0$ and $v^p = 0.3$ are used in the following analyses, besides other parameters. Figure 6 depicts the effect of the pile length on the pile head's longitudinal impedance. It is clear that both the resonance frequency and amplitude of the longitudinal impedance rise with the decrease of pile length in the high-frequency range and this tendency becomes more significant with the rising frequency. In addition, the shorter the pile length, the greater the difference in longitudinal impedance between the two cases with $v^p = 0$ and $v^p = 0.3$. This result demonstrates that

the present solution derived from the Rayleigh–Love model is suitable for the longitudinal vibration of a large-diameter pile, especially when the pile slenderness is low. Furthermore, it can be seen from Figure 7 that the viscous coefficient of soil has an obvious influence on the longitudinal impedance of the pile shaft within the high-frequency range. With the increase of the soil's viscous damping coefficient, both the oscillation amplitude and frequency of the longitudinal impedance decrease.

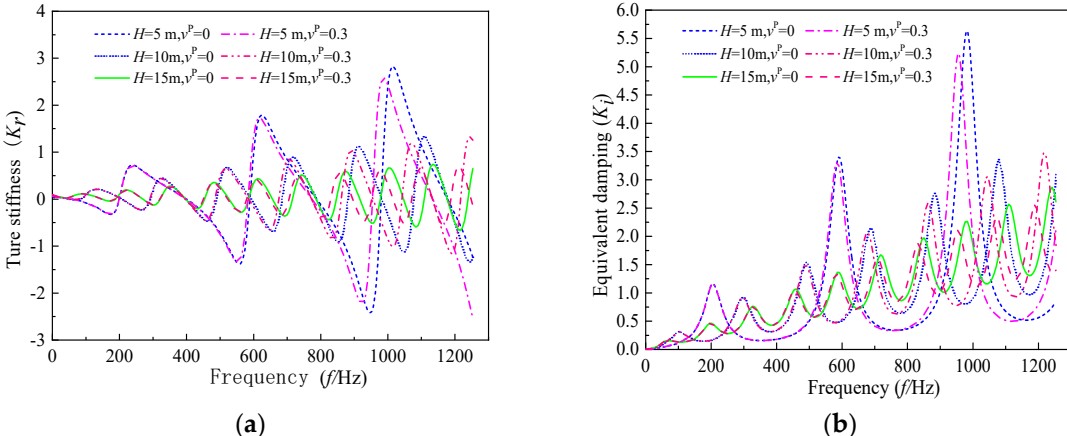

**Figure 6.** The effect of the pile length on the longitudinal impedance of the pile shaft: (**a**) true stiffness; (**b**) equivalent damping.

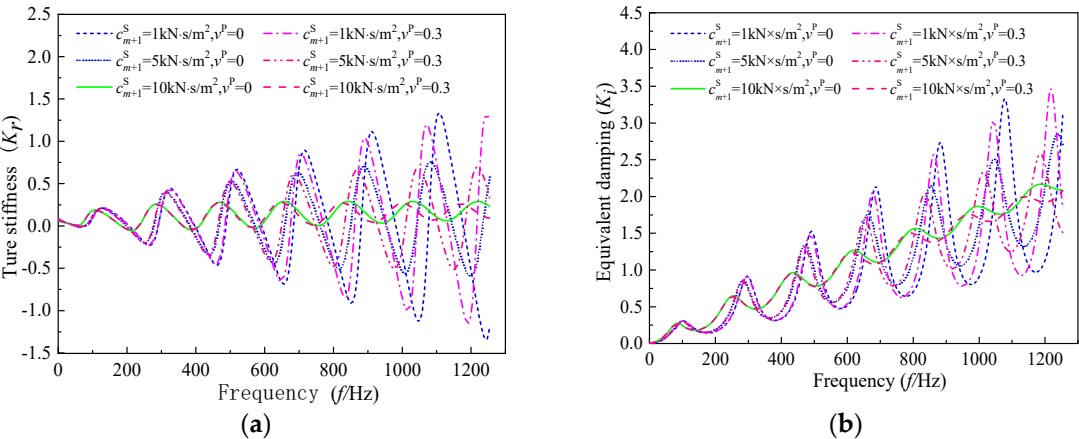

**Figure 7.** The effect of the viscous damping coefficient on the longitudinal impedance of the pile shaft: (**a**) true stiffness; (**b**) equivalent damping.

Figures 8 and 9 depict the effect of the degree and radius soil is weakened due to a construction disturbance on the longitudinal impedance at the head of the pile shaft, respectively. It is clear that the oscillation amplitude and resonance frequency rise with the increase of the degree to which the soil is weakened, and the tendency becomes significant in the high-frequency range. In contrast, only the oscillation amplitude of longitudinal impedance is augmented with the enlargement of the radius of the weakened soil due to a construction disturbance, and the influence of the weakened soil's radius on the resonance frequency is negligible.

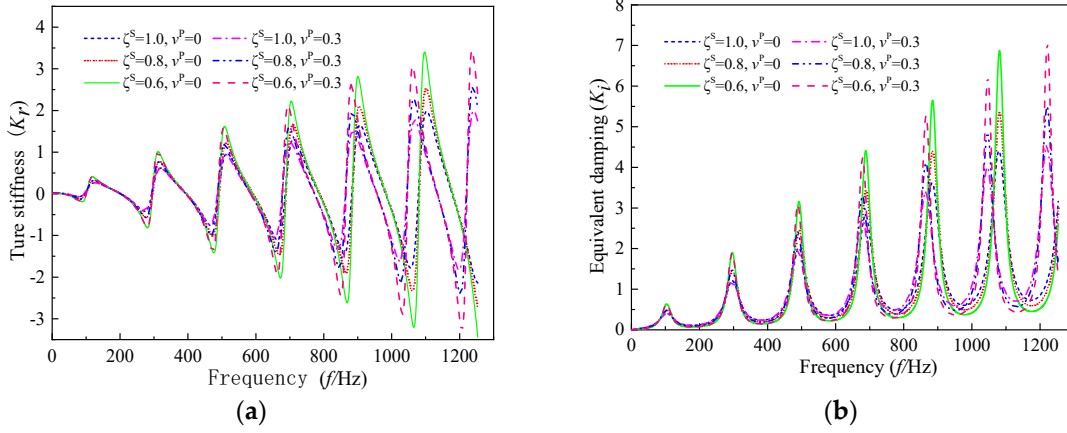

**Figure 8.** The effect of the degree to which soil is weakened due to a construction disturbance on the longitudinal impedance of the pile shaft: (**a**) true stiffness; (**b**) equivalent damping.

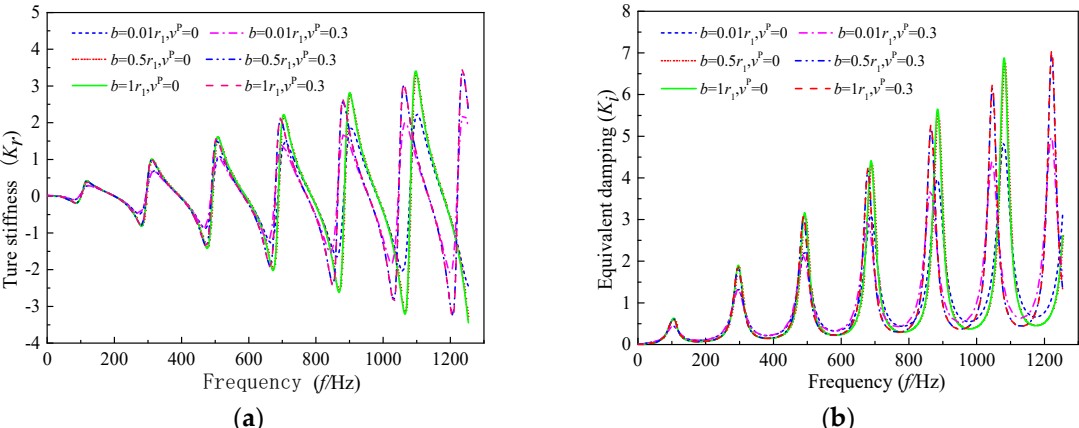

**Figure 9.** The effect of the radius of weakened soil due to a construction disturbance on the longitudinal impedance of the pile shaft: (**a**) true stiffness; (**b**) equivalent damping.

The influences of the degree and radius of strengthened soil on the longitudinal impedance of the pile shaft are illustrated in Figures 10 and 11, respectively. It is observed that the oscillation amplitude and frequency both decrease with the increase of the degree to which the soil is strengthened, which is significant in the high-frequency range. Differently, only the oscillation amplitude of longitudinal impedance becomes smaller with the increase of the strengthened soil's radius due to a construction disturbance, and the influence of the strengthened soil's radius on the resonance frequency can be practically ignored. In addition, the change in the weakened or strengthened soil's radius leads to no further extra effect on the longitudinal impedance when the weakened or strengthened soil's radius reaches a certain value, e.g., $b = 0.5\, r_1$ in this analysis.

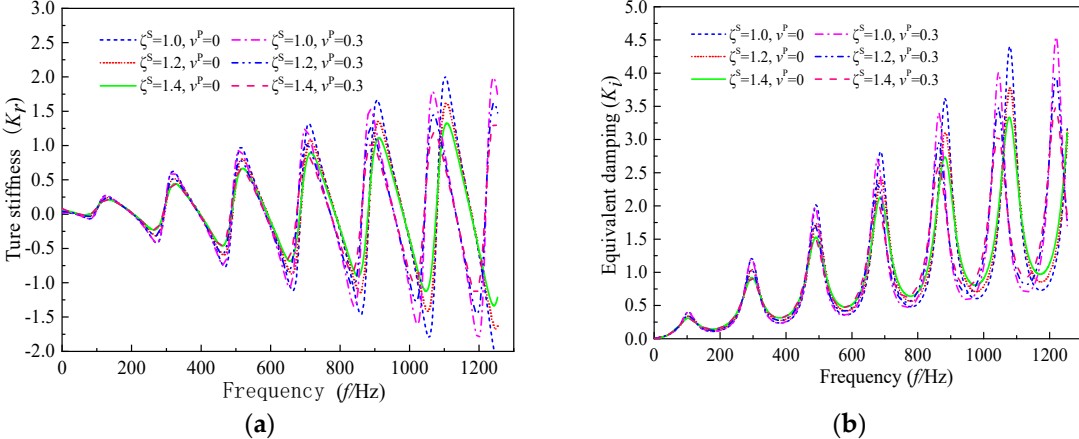

**Figure 10.** The effect of the degree soil is strengthened due to a construction disturbance on the longitudinal impedance of pile shaft: (**a**) true stiffness; (**b**) equivalent damping.

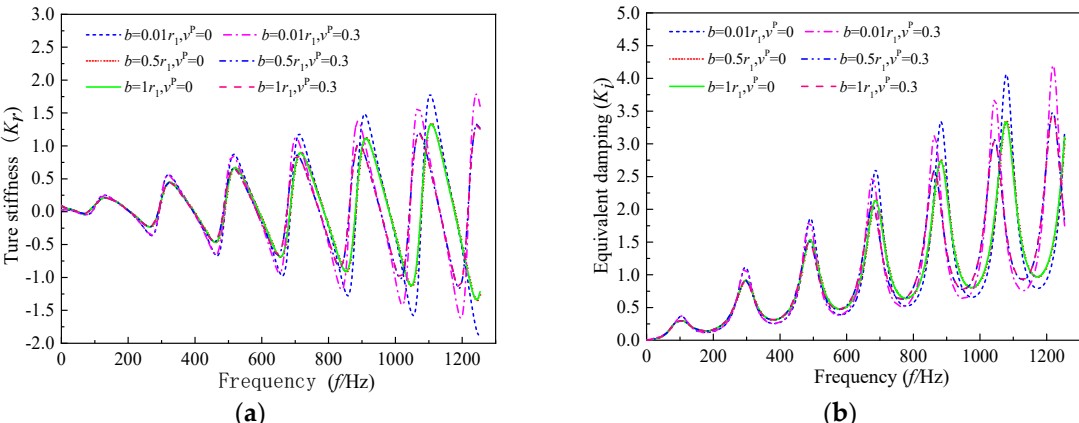

**Figure 11.** The effect of the radius of strengthened soil due to a construction disturbance on the longitudinal impedance of a pile shaft: (**a**) true stiffness; (**b**) equivalent damping.

## 8. Conclusions

A new 3D axisymmetric model was presented to describe the longitudinal vibration of a large-diameter floating pile in viscoelastic surrounding soil with radial inhomogeneity, taking the effects of three-dimensional wave propagation and lateral inertia of the pile shaft into account. The corresponding analytical solution was deduced and validated via comparisons with existing solutions. Parametric analyses were also conducted to examine the influences of Poisson's ratio of the pile shaft, pile length and viscous damping coefficient, as well as the degree and radius of disturbed surrounding soil, on the longitudinal impedance of pile shaft. The relating results demonstrate that:

(1) Both the resonance frequency and amplitude of longitudinal impedance decrease with the increasing Poisson's ratio of the pile shaft in the high-frequency range, while the Poisson's ratio of the pile shaft has a negligible effect on the longitudinal impedance of the pile shaft in the low-frequency range.

(2) The oscillation amplitude and resonance frequency of the longitudinal impedance increase with the decrease of pile length in the high-frequency range and this tendency becomes more significant as the frequency increases. In addition, the shorter the pile length, the greater the difference in the longitudinal impedance.

(3) The viscous damping coefficient of the surrounding soil has an obvious influence on the longitudinal impedance of the pile shaft within the high-frequency range. With the increase of

the viscous damping coefficient of the soil, both the resonance frequency and amplitude of the longitudinal impedance decrease.

(4)	The degree of disturbed surrounding soil has a significant effect on the oscillation amplitude and frequency of longitudinal impedance. In contrast, the change of the radius of disturbed soil has an influence only on the oscillation amplitude, while the influence on the resonance frequency is negligible.

(5)	The presented analytical model and solution are suitable for the longitudinal vibration problem of a large-diameter pile in viscoelastic surrounding soil with radial heterogeneity, especially when the pile slenderness is low. Furthermore, the proposed solution can be easily degenerated to describe the longitudinal vibration problem relating to a large-diameter floating pile in radially homogenous soil or a pile with fixed-end supports.

The pile vibration problem investigated in this manuscript is mainly focused on wave propagation within a pile–soil system where the excitation force intensity is low. The theoretical analysis will overestimate both the stiffness and damping of piles due to the assumption of perfect bonding between the pile and the soil when the excitation force intensity is high.

**Author Contributions:** Conceptualization, Methodology, Software, and Writing—Original draft preparation, K.M.; Supervision, Writing—Review and Editing, Resources, Formal analysis, Project administration, and Funding acquisition, C.C.; Data curation and Investigation, Z.L.; Supervision and Visualization, H.L.; Software and Validation, H.P. All authors have read and agreed to the published version of the manuscript.

**Funding:** This work is supported by the National Natural Science Foundation of China (Grant No. 51878109, 51778107 and 51578100), the Fundamental Research Funds for the Central Universities (Grant No. 3132019601), China's Scholarship Council (CSC No.201806570004), and the Cultivation project of Innovation talent for doctorate students (CXXM2019BS008)

**Acknowledgments:** The corresponding author would like to acknowledge the support from the State Key Laboratory of Coastal and Offshore Engineering, Dalian University of Technology.

**Conflicts of Interest:** The authors declare no conflicts of interest.

## Appendix A

The coefficients of $\gamma'_n$ and $\gamma''_n$ can be written as

$$\gamma'_n = \gamma_n \left[ \frac{1}{\omega/\eta - h^s_{1n}} \sin((\omega/\eta - h^s_{1n})H) + \frac{1}{\omega/\eta + h^s_{1n}} \sin((\omega/\eta + h^s_{1n})H) \right] \tag{A1}$$

$$\gamma''_n = \gamma_n \left[ \frac{1}{\omega/\eta + h^s_{1n}} (\cos((\omega/\eta + h^s_{1n})H) - 1) + \frac{1}{\omega/\eta - h^s_{1n}} (\cos((\omega/\eta - h^s_{1n})H) - 1) \right] \tag{A2}$$

where $\gamma_n$ can be written as

$$\gamma_n = -\frac{(1 + iG'_{1c}\theta)\overline{q}^s_{1n}\overline{\rho}_1\overline{v}^2_1}{\overline{r}_1\left((\overline{h}^s_{1n})^2 - \theta^2\right)\phi^s_n L^s_n} [K_1(\overline{q}^s_{1n}\overline{r}_1) - p^s_{1n}I_1(\overline{q}^s_{1n}\overline{r}_1)] \tag{A3}$$

where $G'_{1c} = c^s_1/(G^s_1 T_c)$, $\overline{h}^s_{1n} = Hh^s_{1n}$, $\overline{q}^s_{1n} = Hq^s_{1n}$, $\theta = \omega T_c$, $T_c = H/\eta$, $\overline{r}_1 = r_1/H$, $\overline{v}_1 = V^s_1/\eta$, $\overline{\rho}_1 = \rho^s_1/\rho^p$, $\phi^s_n$ and $L^s_n$ can be written as

$$\phi^s_n = -p^s_{1n}\left[I_0(q^s_{1n}r_1) - \frac{2\pi r_1 q^s_{1n}}{\rho^p A^p} \frac{(G^s_1 + c^s_1 s)}{(\eta h^s_{1n})^2 - \omega^2} I_1(q^s_{1n}r_1)\right] + \left[K_0(q^s_{1n}r_1) + \frac{2\pi r_1 q^s_{1n}}{\rho^p A^p} \frac{(G^s_1 + c^s_1 s)}{(\eta h^s_{1n})^2 - \omega^2} K_1(q^s_{1n}r_1)\right] \tag{A4}$$

$$L^s_n = \int_0^H \cos^2(h^s_{1n}z)dz \tag{A5}$$

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
