# Peer review of "An Analytical Solution for Longitudinal Impedance of a Large-Diameter Floating Pile in Soil with Radial Heterogeneity and Viscous-Type Damping"

_applsci, doi:10.3390/app10144906_

Round 1
Reviewer 1 Report
This is very interesting work on the analytical solution of a pile in Heterogenous soil. Authors efforts are appreciable for such a complex problem. I would argue too complex to have an analytical solution. However, the authors' effort will help but it should discuss the limitation of this analytical solution in the introduction. Apart from this, I have two major concern-
- Given the mathematical formulations are correct ( I am not an expert), the analysis must have been done via a program. The mathematical implementation was not clear to me. I tried to follow with notations, some of them are not clear or explained. What is inner zone length and how do you define. I believe, authors should supply their program/calculation as a supplemental file to check. Readers can check /use it as well, otherwise what is the use of such a complicated solution.
- Authors should put an effort to compare their observation with experimental or real pile data. I am not saying they should have their data, but something from literature.
editorial correction:
Improve Fig. 1; the symbols and notations are not clear. This is important to understand the model.
Describe all parameters in the axis title of the figures.
After conclusion, you may describe the limitation of this work.
Author Response
Point 1: Given the mathematical formulations are correct ( I am not an expert), the analysis must have been done via a program. The mathematical implementation was not clear to me. I tried to follow with notations, some of them are not clear or explained. What is inner zone length and how do you define. I believe, authors should supply their program/calculation as a supplemental file to check. Readers can check /use it as well, otherwise what is the use of such a complicated solution.
Response 1:
- To improve the readability of the mathematical implementation, we have added a diagram for the derivation procedure in the revised manuscript.
- 1 has been improved to make the symbols and notations clearer.
- The construction operations generally disturb the soil in the immediate vicinity around the pile, that is, the pile shaft is surrounded by an annular disturbed zone. In this paper, the ‘inner zone length’ denotes the range of disturbed zone, which is defined by the radial thickness of the inner disturbed zone. We have added the notation of inner zone length in Fig.1.
- The program saved as ‘.m’ file was built by MATALAB 2017b version. We have uploaded our program as a supplemental file.
Point 2: Authors should put an effort to compare their observation with experimental or real pile data. I am not saying they should have their data, but something from literature.
Response 2: Existing researches for the analytical method of pile vertical vibration commonly adopt comparison with the previous mature solution to verify the accuracy of their proposed solution[1-5]. The accuracy of our analytical solution can be validated with independent comparisons shown in the section of ‘ 7.1 Verification of the solution’ [3, 6]. However, the reviewer's suggestion is still advisable, we plan to further compare our observation with experimental or real pile data in the future work by using numerical method e.g. FEM.
[1] Z. Y. Li, K. H. Wang, W. B. Wu, C. J. Leo, N. Wang, Vertical vibration of a large-diameter pipe pile considering the radial inhomogeneity of soil caused by the construction disturbance effect, Comput. Geotech. 85(2017) 90-102. https://doi.org/10.1016/j.compgeo.2016.12.016
[2] C. J. Zheng, X. M. Ding, Y. F. Sun, Vertical vibration of a pipe pile in viscoelastic soil considering the three-dimensional wave effect of soil, Int. J. Geomech. 16(2016) 04015037. https://doi.org/10.1061/(asce)gm.1943-5622.0000529
[3] S. H. Lu, K. H. Wang, W. B. Wu, C. J. Leo, Longitudinal vibration of a pile embedded in layered soil considering the transverse inertia effect of pile, Comput. Geotech. 62(2014) 90-99. https://doi.org/10.1016/j.compgeo.2014.06.015
[4] W. B. Wu, G. S. Jiang, B. Dou, C. J. Leo, Vertical dynamic impedance of tapered pile considering compacting effect, Math. Probl. Eng. 2013 1-9. https://doi.org/10.1155/2013/304856
[5] W. B. Wu, K. H. Wang, Z. Q. Zhang, C. J. Leo, Soil‐pile interaction in the pile vertical vibration considering true three‐dimensional wave effect of soil, Int. J. Numer. Anal Met. 37(2013) 2860-2876. https://doi.org/10.1002/nag.2164
[6] Yang, D.Y.; Wang, K.H.; Zhang, Z.; Leo, C.J. Vertical dynamic response of pile in a radially heterogeneous soil layer. Int. J. Numer. Anal. Met. 2009, 33, 1039-1054. doi: 10.1002/nag.755.
Editorial correction:
Point 1: Improve Fig. 1; the symbols and notations are not clear. This is important to understand the model.
Response 1: We have modified Fig.1 in the revised manuscript.
Point 2: Describe all parameters in the axis title of the figures.
Response 2: We have added description for all parameters in the axis title of the figures in the revised manuscript.
Point 3: After conclusion, you may describe the limitation of this work.
Response 3: We have added a description of the limitation for this work after conclusion in the revised manuscript. “The pile vibration problem investigated in this manuscript is mainly focused on the wave propagation within the pile-soil system where the excitation force intensity is low. The theoretical analysis will overestimate both the stiffness and damping of piles due to the assumption of perfect bonding between pile and soil when the excitation force intensity is high.” Please check the corresponding content.

Reviewer 2 Report
ID: applsci-854663
Title: An analytical solution for longitudinal impedance of a large-diameter floating pile in soil with radial heterogeneity and viscous-type damping
This paper proposed a new analytical model for longitudinal impedance of a large-diameter floating pile in visco-elastic soil with radial heterogeneity and viscous-type damping. The aim of the proposed analytical solution is to handle the longitudinal vibration of the large-diameter floating pile taking the effect of 3D wave propagation and lateral inertia of the pile shaft. Parametric study are also conducted to investigate the longitudinal vibration characteristics of the pile.
However, there are several missing parts related to equations derivation. Most part of this paper is derivation of the equation. But, several connections from equation to equation are missing.
Line 86: what is the mechanical constants?
Line 103 & 107: Please provide reference information for the governing equations.
Line 106: EQ2 and EQ3 seems to be the same. Please check it.
Line 109: Section 4. Boundary and Initial Conditions
Please provide a detailed information related to Eq 5b, Eq7b, and Eq 8b, such as how to derive, where it comes from, etc.
Minor :
Line 80: second “i.e.” term can be removed.
Line 83&84: there are five parameters. But only four items are denoted. Please check them.
Author Response
Point 1: Line 86: what is the mechanical constants ?
Response 1: To properly consider the vibration characteristic of floating pile, the boundary condition beneath pile toe is simplified as the Voigt model [1-3]. The mechanical constants and refer to the spring constants and damping coefficients of the Voigt model, respectively.
[1] L. Gao, K. H. Wang, S. Xiao, Z. Y. Li, J. T. Wu, An analytical solution for excited pile vibrations with variable section impedance in the time domain and its engineering application, Comput. Geotech. 73(2016) 170-178. https://doi.org/10.1016/j.compgeo.2015.12.008
[2] C. Y. Cui, S. P. Zhang, C. David, K. Meng. Dynamic impedance of a floating pile embedded in poro-visco-elastic soils subjected to longitudinal harmonic loads, Geomech. Eng. 15(2018) 793-803. https://doi.org/10.12989/gae.2018.15.2.793
[3] X. M. Ding, H. L. Liu, C. Jian, K. Cheng, Time-domain solution for transient dynamic response of a large-diameter thin-walled pipe pile, Earthq. Eng. Eng. Vib. 14(2015) 239-251. https://doi.org/10.1007/s11803-015-0020-7
Point 2: Line 103 & 107: Please provide reference information for the governing equations.
Response 2: We have given corresponding reference information for the governing equations in revised manuscript.
- Line 103: Based on the wave propagation theory of continuum in the axisymmetric condition, the governing equation for the jth disturbed sub-layer proposed by Nogami and Novak [34] is adopted.
- Line 107: According to the Rayleigh–Love rod model theory [31], the governing equation for the longitudinal vibration of large-diameter pile shaft can be written as.
Point 3: Line 106: EQ2 and EQ3 seems to be the same. Please check it.
Response 3: We have checked it and replaced Eq.3 with .
Point 4: Please provide a detailed information related to Eq 5b, Eq7b, and Eq 8b, such as how to derive, where it comes from, etc.
Response 4: We have added detailed information related to Eq 5b, Eq7b, and Eq 8b in the revised manuscript.
- 5b: where refers to the internal stress of soil. and are the external forces related to stiffness and damping, respectively. According to the force equilibrium conditions, Eq.(5b) can be established.
- 7b: where and are the shear stress of the jth and j+1 th, respectively, sub-layers. Based on the stress equilibrium condition at the interface between jth and (j+1) th sub-layers, Eq.7b is built.
- 8b: where denotes the external supporting force beneath pile toe. is the internal stress of pile. According to the force equilibrium condition beneath pile toe, Eq.(8b) can be established.
Point 5: Line 80: second “i.e.” term can be removed.
Response 5: We have removed the abovementioned "i.e." in the revised manuscript.
Point 6: Line 83&84: there are five parameters. But only four items are denoted. Please check them.
Response 6: We have checked them and added a definition for “ ” in the revised manuscript. Please check the corresponding content.

Reviewer 3 Report
The paper presents an analytical model for solving the longitudinal impedance of a large‐diameter floating pile, which is interesting for publishing.
There are 39 references in the text, being about 40% from the last 5 years and about 47% are more than 10 years old, which is acceptable.
I found the paper well organized and conclusions are interesting. The adopted models are robust, despite only considering the linear response of the soil, which means that the solution is only valid for problems that do not involve high vibration levels. Therefore, I recommend adding a brief paragraph highlighting this issue. Another issue is related to the values adopted for the mechanical constants of visco‐elastic supports (in lines 164 and 165), which are both equal to 1e5. This seems quite odd to me, so the authors should clarify how these values were chosen.
Figures 5, 6, 7 and 8 are presented before they are properly referenced in the text, which should be corrected.
I recommend that the paper should be accepted for publishing after minor revision.
Author Response
Point 1: The adopted models are robust, despite only considering the linear response of the soil, which means that the solution is only valid for problems that do not involve high vibration levels. Therefore, I recommend adding a brief paragraph highlighting this issue.
Response 1: We have added a description of the limitation for this work after conclusion in the revised manuscript. “The pile vibration problem investigated in this manuscript is mainly focused on the wave propagation within the pile-soil system where the excitation force intensity is low. The theoretical analysis will overestimate both the stiffness and damping of piles due to the assumption of perfect bonding between pile and soil when the excitation force intensity is high.” Please check the corresponding content.
Point 2: Another issue is related to the values adopted for the mechanical constants of visco‐elastic supports (in lines 164 and 165), which are both equal to 1e5. This seems quite odd to me, so the authors should clarify how these values were chosen.
Response 2: In this paper, the corresponding spring constants and damping coefficients in the following analysis are given values of and , respectively, which are referred to relevant existing researches [1-3].
[1] Lü, S.H.; Wang, K.H.; Wu, W.B.; Leo, C.J. Longitudinal vibration of a pile embedded in layered soil considering the transverse inertia effect of pile. Comput. Geotech. 2014, 62, 90-99. doi:10.1016/j.compgeo.2014.06.015.
[2] Ding Xuan-ming, Zheng Chang-jie, Liu Han-long. A theoretical analysis of vertical dynamic response of large-diameter pipe piles in layered soil[J]. Journal of Central South University, 2014, 21(8):3327-3337.
[3] Li Z, Wang K, Wu W, et al. Vertical vibration of a large-diameter pipe pile considering the radial inhomogeneity of soil caused by the construction disturbance effect[J]. Computers & Geotechnics, 2017:90–102.
Point 3: Figures 5, 6, 7 and 8 are presented before they are properly referenced in the text, which should be corrected.
Response 3: As per your suggestion, we have corrected it in the revised manuscript. Please check the corresponding content.

Round 2
Reviewer 1 Report
it seems they addressed the majority of my concerns. this can be published now.
Reviewer 2 Report
Good jobs.